# Vulnerabilities of Rohingya and host community children in Bangladesh: A qualitative study on child labor, well-being, and the impact of COVID-19

Ahmed Hossain[1,2]*, Mohammad Ali[3], Heba Hijazi[1,4], Mohamad Alameddine[1]

**1** College of Health Sciences, University of Sharjah, Sharjah, United Arab Emirates, **2** Department of Public Health, North South University, Dhaka, Bangladesh, **3** School of Allied Health, La Trobe University, Melbourne, Victoria, Australia, **4** Department of Health Management and Policy, Faculty of Medicine, Jordan University of Science and Technology, Irbid, Jordan

* ahossain@sharjah.ac.ae

## Abstract

Stateless Rohingya children in Bangladesh's refugee camps, lacking citizenship or legal safeguards, face heightened risks of human rights abuses like child labor and early marriage. This qualitative study explored these challenges, comparing Rohingya and host community children, specifically regarding child labor and well-being. Conducted from February 23 to March 18, 2022, the research employed 20 in-depth interviews with children (aged 11–17) and 20 Key Informant Interviews (KIIs) with stakeholders and leaders. Due to restrictions, child interviews relied on note-taking, while KIIs were audio-recorded, transcribed, and thematically analyzed. A combination of convenience and purposive sampling methods was used to select participants. Findings revealed that child labor significantly harms well-being in both communities. Children's vulnerability makes them easy targets for exploitation through low wages and intimidation. Economic necessity frequently compels families to send children to work, severely impacting their education and development. Older boys (15–17 years) were more commonly engaged in paid work than girls. The COVID-19 pandemic exacerbated the situation, increasing child labor in sectors like shop/restaurant assistance and driving three-wheelers. The KIIs identify specific types of labor, such as carrying goods, farming, and construction work, that were prevalent among children. Key drivers included illness of primary earners, lack of access to education, and children needing income for personal expenses. Hazardous tasks like handling gas cylinders or carrying heavy loads pose serious health risks, causing physical pain, injuries, and mental strain. KIIs further highlighted rampant camp issues like child marriage, trafficking, and other violence, disproportionately affecting girls. Urgent, comprehensive interventions are critical. Breaking this cycle requires collaborative stakeholder efforts to establish robust support systems

**Data availability statement:** The details about the design, settings, protocol and data are available at https://figshare.com/account/home#/projects/233549.

**Funding:** The author(s) received no specific funding for this work.

**Competing interests:** The authors have declared that no competing interests exist.

encompassing education, healthcare, and legal protections. Without immediate action, these children's futures remain perilously at risk, perpetuating cycles of poverty and exploitation.

## Introduction

Child labor in low- and middle-income countries adversely affects children's physical and mental health [1]. These impacts include poor growth, malnutrition, higher rates of infectious diseases, injuries, behavioral and emotional disorders, decreased coping efficacy, and increased risk of trauma [1]. Child labor remains a pervasive issue in Bangladesh, particularly among vulnerable populations such as the Rohingya refugees and host communities in Bangladesh [2,3]. The socioeconomic drivers of child labor are complex and multifaceted, rooted in systemic poverty, lack of educational opportunities, and the aftermath of crises such as the COVID-19 pandemic [4–6]. The vulnerabilities faced by Rohingya and host community children in Bangladesh have become increasingly pronounced, particularly in the context of child labor and the impacts of the COVID-19 pandemic. This qualitative study explores the multifaceted issues surrounding child labor, well-being, and the exacerbating effects of the pandemic on these vulnerable populations.

Cox's Bazar, Bangladesh, hosts the world's largest refugee camp, housing over a million Rohingya refugees fleeing persecution in Myanmar [7–9]. This influx has strained resources, leading to overcrowding, limited healthcare, and socioeconomic challenges for both refugees and host communities [8–10]. Currently, approximately 300,000 children aged 5–17 years reside in the Rohingya refugee camps, accounting for 68.5% of the total Rohingya child population (ages 0–17) [11,12]. Child protection issues like child labor, child marriage and school dropouts are prevalent in this community [12].

Poverty remains the most significant factor driving child labor, as families facing extreme financial hardship are often compelled to depend on their children's income to meet basic survival needs such as food, shelter, and healthcare [12–14]. In regions like Cox's Bazar, where both refugee and host communities struggle with limited resources, children are frequently forced into work to contribute to their household's income. This creates a cycle where education is deprioritized, and children are exposed to unsafe and exploitative labor conditions, further entrenching their families in poverty.

In the realm of health research, child labor is recognized as a critical public health concern. Studies consistently show that children engaged in labor are exposed to hazardous conditions that result in poor growth, malnutrition, injuries, and long-term health consequences such as chronic pain or respiratory illnesses [1]. Mental health impacts, including behavioral disorders and emotional distress, are also prevalent but less studied due to measurement challenges [15]. These adverse outcomes not only affect children during their working years but also have lasting effects into adulthood.

The COVID-19 pandemic during the time 2021–22 dramatically worsened these existing vulnerabilities [16]. Lockdowns and movement restrictions, along with the

global economic downturn, have resulted in significant job losses for adults, driving an even greater reliance on child labor in many affected families [1,17]. School closures during the pandemic have also disrupted children's education, pushing them further into the workforce. In addition, the pandemic has strained access to essential services such as healthcare, nutrition, and social protection, leaving children without the support systems needed to break free from child labor. Consequently, the pandemic has not only increased the prevalence of child labor but has also heightened the long-term risks to children's health, education, and overall well-being.

Addressing child labor in refugee and host communities in Bangladesh requires a comprehensive understanding of the underlying socioeconomic dynamics. This qualitative study explores the experiences of children from both the Rohingya and host communities, with a focus on their involvement in child labor, overall well-being, and the effects of the COVID-19 pandemic on their lives. By understanding the key factors driving child labor and its impact, the study aims to guide evidence-based interventions that will protect children and enhance their well-being.

## Methods and materials

### Ethics statement

We took the ethical approval from the North-South University institutional review board (IRB number: 2022/OR-NSU/IRB/0103). Written consent was also obtained from the RRRC for conducting interviews with the Rohingya refugees (RRRC/NGO/Research Work/ 1–3 120201 4407). Respondents' participation in the study was entirely voluntary, and informed consent was obtained from each participant. Prior to participation, the underage participant's available guardian, either their parents or the business owner, verbally agreed to their involvement in a note taking interview. The decision to involve business owners as guardians likely stems from logistical challenges in reaching parents. Business owners often exert considerable control over children's time and availability, particularly in cultures or situations where children working under them are viewed as their responsibility during working hours. Additionally, some children may live far from their parents, work in different locations, or have limited communication with their families. In such cases, obtaining consent from the business owner, who has immediate oversight, becomes a more practical solution. Data collectors were trained to treat all respondents with respect, be fair and honest, and avoid applying any pressure on the respondents. We protected individuals' confidentiality following their participation in the study. Therefore, all procedures performed in studies involving participants followed the ethical standards of the NSU research committee and the 1964 Helsinki Declaration and its later amendments or comparable ethical standards.

### Study design and study settings

This formative research was conducted in Cox's Bazar District of Bangladesh from February 23, 2022, to March 18, 2022. The study regions were divided into two communities: Rohingya and host communities. Participants from the Rohingya community were selected from Ukhiya and Teknaf sub-districts, the primary hosts of refugee settlements since the 2017 influx. These areas face direct impacts on infrastructure and public services. In contrast, informants from the host community were chosen from Ramu and Chakaria, which, while not hosting refugees directly, represent communities indirectly affected by the crisis. This selection provided a comprehensive understanding of the interactions and impacts between refugees and host populations across different levels of proximity.

The study included two distinct participant groups: 20 children and 20 key informants, comprising stakeholders and local leaders. To ensure a comprehensive understanding of the topic, both in-depth interviews and Key Informant Interviews (KIIs) were conducted. This dual approach captured the lived experiences of children while incorporating the perspectives of influential community members.

A combination of convenience and purposive sampling was utilized to identify participants. Convenience sampling allowed researchers to engage participants who were readily accessible, while purposive sampling ensured the inclusion of individuals with specific knowledge or insights relevant to the study.

## Child labor

According to the ILO, child labor is "work that exceeds a minimum number of hours, depending on a child's age and the type of employment." Children in work are divided into three categories by the ILO: economically active children, child labor, and hazardous work. If a child works for at least one hour outside of school or at home once every seven days, they are considered economically active. If a child is under the age of 18 or is doing a hazardous job, they are considered doing child labor. If children engage in activities that may impair their physical, mental, or developmental health or safety, they are classified as performing hazardous work. Within the legal frameworks of Bangladesh, child labor is defined by the Bangladesh Labour Act, 2006, as the employment of children under 14 years of age, which is strictly prohibited, while hazardous forms of work are banned for anyone under 18 years [18].

## Sampling technique and sample size

In this process, 20 in-depth interviews and 20 key-informant interviews (KIIs) were conducted. The in-depth interviews with children were taken in different marketplaces. The distribution of KIIs is given in Table 1. The 8 informants from the Rohingya community and 5 informants from the host community were interviewed at the convenience of the data collectors based on their interest in participating and the availability of the informants at the time of visit to their households. The 7 key informants were selected purposively from RRRC and NGO professionals so that programmatic initiatives, community approaches, and gaps in the system could be properly captured.

## Data collection

In the qualitative study, we performed twenty in-depth interviews (IDIs) and twenty KIIs. For IDIs, working children were interviewed, while for KIIs, community leaders and NGO personnel who are involved in child protection were interviewed. The KII participants were encouraged to elaborate and describe their experiences and viewpoints on child labor in-depth during the interviews, which were based on open-ended questions. Before the KII interviews, the researcher asked for permission to record the interviews and took handwritten notes. These interviews lasted approximately 20–30 minutes, or however long a participant may choose to talk. Audio-recorded interviews were translated as required and transcribed (using https://otranscribe.com) by the researcher or a trained transcriptionist who signed a confidentiality agreement. The questionnaires for IDI and KII are provided in the supporting information S1 Text.

## Data analysis

Due to the restrictions on recording the statements provided by the children, the study adopted the note-taking format to derive the information from them. We used a thematic analysis approach to outline, describe, and report the key patterns within and across the theme-wise responses. All data were organized under various subheadings or themes that aligned with the research objectives, such as: the types of labor children were primarily involved in, the causes of child labor (parental attitudes towards sending their children to work, vulnerability after a man-made disaster, and child labor due to

**Table 1. Distribution of Key Informant interviews (KIIs).**

| Identity | Number (n=20) | Gender Distribution | Roles |
|---|---|---|---|
| Refugee Community | 8 | M: 5, F: 3 | Majhi: 2 (M), Teacher: 2 (M)<br>Imam: 1 (M), Common People: 4 (M:1, F:3) |
| Host Community | 5 | M:3, F:2 | Imam & Community Leader: 1 (M), Common people: 4 (M:2, F:2) |
| RRRC | 1 | M:1 | Administrative personnel: 1 (M) |
| NGO professionals | 6 | M:4, F:2 | Child-protection experts: 6 (M:4, F:2) |

the COVID-19 pandemic), the involvement of children in hazardous work, gender segregation, challenges the children faced and their consequences on their development and wellbeing. In many cases, verbatim or direct quotations from the respondents were included in the data analysis.

### Language

A language expert did the translation from English to Bangla. The reverse translation (from Bangla to English) was also done by a medical professional who speaks Bangla as a first language. The questionnaire was not translated into the Rohingya dialect. Though the accents of the languages differ, Bangla is the most widely utilized language in Rohingya settings for communication. The researcher predicted that a few of the respondents would not be able to provide the information requested because they did not comprehend Bangla to the requisite degree. To help alleviate the situation, we hired local enumerators who had received professional training in surveying both Bangla and Rohingya.

## Results

### Note-taking from interviewed children

Twenty interviews were conducted with working children from the Rohingya and host communities who were employed in different marketplaces. The age of children who were selected to participate in this study ranged from 14 to 17 years old. They were involved in diverse jobs, such as assisting in shops and hotels, driving vehicles, etc. The respondents had already been engaged in labor for 3 weeks to 4.5 years at the time of the interview.

A majority of children interviewed (5 out of 10 from the host community) stated that their fathers played a key role in sending them to paid work. Fathers from the host community, as primary breadwinners in many traditional family structures, may feel compelled to involve their children in earning an income to support the household. This highlights the fathers' influence over family decisions related to financial matters. The father's death or a parent's ailment was another reported reason for such an engagement. In almost all cases, poverty was the main driver for forcing children to assume the role of wage earners in the household. Furthermore, the host children cited poverty and lack of time due to engagement in labor as the main reasons for dropping out of school. A few host children revealed that they face workplace violence (both verbal and physical) from their employers.

On the other hand, in the Rohingya camps, the ages of the interviewed children engaged in paid labor ranged from 11 to 18 years. At the time of the interview, these children had already been engaged in labor for 4 weeks to 2 years. It is culturally acceptable or expected for fathers from Rohingya community to make decisions about their family and children's work, especially when financial needs are pressing. Many of these children (8 out of 10 children) either jointly decided with their parents to get involved in paid work outside of the home or were sent by their mother. A few children reported that they started working to sustain the family following the ailment of the wage-earner in the family throughout the ongoing COVID-19 pandemic. Many said that they stopped going to school as they had completed level-4 education and there were no provisions to study any further. Further, a few cited the unavailability of teaching staff as being responsible for children's dropping out of school. The children principally mentioned two problems in their working places: painful, tedious work and untimely, partial, or no payment. Children assigned tasks such as carrying heavy items, construction work, or repetitive jobs like restaurant work and selling goods face significant physical and mental strain. These activities offer little personal growth and may lead to long-term health issues. Additionally, being underpaid or unpaid highlights exploitation and deepens their families' economic hardships, perpetuating the cycle of poverty.

### Impact of COVID-19 on child labor dynamics from KIIs

When asked about how the COVID-19 pandemic impacted host children's engagement in labor, the respondents said that no child could work outside the home for payment during the lockdowns and restrictions. Moreover, as employment opportunities shrank, the number of working host children also decreased. A similar trend was observed in the Rohingya camps,

as mentioned by the informants. However, a few participants in this study revealed a riveting aspect of the refugee camps: children were still getting involved in distributing aid, support, or sustenance materials during the lockdowns for money, which included carrying heavy-weight packages from one place to another. An NGO respondent added,

*Schools got closed in COVID-19, and children who used to study before began to engage in child labor.*

From the participant's perspective, although the pandemic did not increase child labor, it increased trauma and disputes among children, which at times stretched to their parents. On the other hand, due to COVID-19 restrictions, the refugee camps lost revenue, which led to a drop in receiving relief or special assistance to the Rohingya families. Participants mentioned that to mitigate this gap, families were more inclined towards making the male child or adolescent go to work for whatever they could earn to maintain the household. Additionally, they said that child marriage and polygamy tended to increase as coping strategies to alleviate the shortage of food. The greater the number of new members, the greater the number of ration cards that would be added for food supply and assistance. Furthermore, after the pandemic restrictions were lifted, children were more frequently engaged in work such as assisting in shops and restaurants and driving three-wheelers.

### Types and socioeconomic drivers of child labor from KIIs

The stakeholders were asked about the types of work that the children were primarily doing in both communities. According to the host participants, children between 5 and 12 years of age did not usually work. Instead, they went to school and were frequently observed among the girl children. By contrast, they mentioned that children of 13–18-years-old commonly work and are generally involved as welders or electricians on construction sites, in the dry fish processing industry, as carpenters, or as shop assistants in groceries and pharmacies. Moreover, they were inclined to get involved in seasonal cultivation and fishing. Interestingly, the respondents also revealed that the host children tended to conduct business in association with the children from the Rohingya refugee camps. On the other hand, the camp children were usually involved in a wider range of labor to earn money, such as carrying heavy items, for instance, gas cylinders, construction materials, rice sacs, other relief or hygiene kits, carrying grocery bags, etc., growing vegetables, catching fish and selling them as well as ice cream, helping to construct shelters, etc. One of the key informants from the Rohingya camp mentioned:

*Adults inside the camp make children carry gas cylinders and pay 5–10 taka for that only.*

Additionally, the informants mentioned that the Rohingya children were also seen working outside the camps as drivers or helpers of a battery-operated rickshaw, which is locally called a TomTom and is owned by the host dwellers. This type of work poses significant hazards for these Rohingya children. They were also frequently seen working as assistants in tea stalls, grocery shops, restaurants, and pharmacies, according to them.

### Involvement of children in hazardous work from KIIs

As mentioned in the previous section, the host children often get involved in hazardous activities like welding, electrical work, carpentry, etc., and the key people of the community said the main reason might be poverty or a lack of interest in going to school. Conversely, they identified the risky work that is usually done by the Rohingya children, such as carrying heavy weights (e.g., gas cylinders, construction materials, rice sacks, other relief or hygiene kits, heavy grocery bags, etc.), which is painful and has a high probability of having deleterious effects on their health in the long run. The participants believed that driving vehicles was hazardous for the Rohingya children. A child-protection expert from an NGO stated:

*Children of 8–12 years of age carry 15–20 kgs, which can damage their spinal cord and also hinder physiological development, but their parents do not understand this. Rather, they prioritize the lump sum amount their children earn.*

## Health and well-being

Children engaged in labor face numerous health risks associated with their work conditions. Many reported experiencing physical pain due to the strenuous nature of their jobs, which can have long-term health implications. The types of work vary significantly between communities; while host community children may be involved in hazardous occupations such as welding or electrical work, Rohingya children often perform physically demanding tasks like carrying heavy items or assisting in construction. One NGO worker from the Rohingya community stated

> *Many children were observed in the market engaged in hazardous work under poor conditions, putting their health, safety, and moral development at risk.*

Children engaged in tasks such as carrying heavy loads, construction work, or repetitive jobs like restaurant work and selling goods experience both physical and mental strain. These physically demanding activities can lead to injuries and long-term health problems, while repetitive tasks may cause mental fatigue and stress. Additionally, such jobs provide minimal opportunities for personal development or skill acquisition, hindering their growth and future prospects. This type of labor not only compromises their health but also perpetuates cycles of poverty by depriving them of education and meaningful learning experiences.

## Parents' attitude towards sending their child to labor from KIIs

The respondents from the Rohingya community indicated that these children were commonly forced by their families to do such work and asked to bear their living expenses with their own earnings. Furthermore, they perceived that by staying and working in this community, their parents did not tend to see child labor as something detrimental but rather as a regular and long-standing practice. They emphasized that there is a notion of single mothers and households with unemployed fathers in the camps sending their male children, mostly 11–17 years of age, to work.

Additionally, it was mentioned that some mothers have a compulsion to send their children to labor against their will. On the other hand, the host children are frequently sent to work by their father. However, no such contradiction was reported by the respondents of the host community. In this regard, one of the informants from the Rohingya community stated:

> *In female-headed households, where a father or elder son is absent, mothers send their 10–12 years old child to work and earn. In summer, such single mothers send their children to sell ice cream where they want to join learning sessions, and sometimes they get beaten by their mother if they do not want to sell ice cream.*

## Gender segregation in child labor from KIIs

Based on the informants' statements, the host girl children were usually not involved in labor outside the household. They generally either went to school or did household chores. However, they also mentioned that the girls aged 16–18 years were usually involved with local NGO work, whereas the boys rarely received such opportunities.

On the other hand, the informants from the Rohingya community said that the migrant minorities were very protective of their girl children. They had a high propensity to impose restrictions on their movement outside the home when they reached a certain age. In this context, one of the participants from the camp said:

> *Rohingya are sensitive regarding their girl children, so they prefer to keep them behind a curtain.*

It was understood from the interview that the girls of the Rohingya community frequently get married at 14–15 years of age. The adolescent girls primarily babysat and assisted their mothers with housework.

 

### Reasons for child labor predominance from KIIs

In both the host and Rohingya communities, the majority of the informants narrated that it was easier for children to be employed than adults, as they could be persuaded with negligible salaries and could easily be kept under control with intimidation. Additionally, because of not being educated, children did not demand justifiable remuneration for their intense labor. During the interview, a respondent added this context:

> *Children are satisfied with little payment, and they do not demand much.*

One informant mentioned that the Rohingya children work up to 12–16 hours daily for as little as one-fourth of the salary an adult would draw for the same work.

### Challenges faced by the children involved in labor from KIIs

While investigating the challenges faced by the working children of both the host and Rohingya communities, an opposite trend was observed. They narrated that adolescent boys from their community often got involved in the prohibited illicit substances (e.g., Yaba, etc.) trade orchestrated by criminal groups in the camps. As highlighted by one of the host community informants:

> *Children from our community get involved in the illegal trade in Yaba tablets in the Rohingya camps.*

According to their statement, only vigilant host parents would seek help from the community leaders and local law enforcement to rescue their children from bad company. They believed that there was no risk in reporting such events and that the parents had the liberty to do so to the Union Parishad chairman and the members. However, it was comforting to know from two participants that host parents had an increasing tendency to counsel their children through religious leaders of both communities to avoid mixing with bad company. A respondent from the Rohingya community said that the children repeatedly got involved with armed rebel groups and were recruited at an early age, as they were easy to manipulate. He also mentioned an increase in Rohingya child trafficking, as they frequently aspire to leave the camps.

### Challenges in reporting the child protection issues from KIIs

The respondents, who were child protection specialists in different organizations at the Rohingya camps, said that the agencies were working to outreach and respond to child protection issues as a whole. They believed focus should be given to issues like child marriage, child labor, human trafficking, violence and abuse, etc., reporting and the referral pathways individually. However, most informants identified a lack of confidentiality within the Rohingya community as a significant problem while reporting such legal protection issues. Although the readiness to report was noticed, the awareness of the need to do so was low, as most of the study participants stated. Additionally, they mentioned that they suspected a looming threat of retaliation from armed rebel groups, for which parents were not inclined to report these problems. To quote one informant:

> *It is easier for armed rebel groups to recruit children, and children are at risk of joining them, children are also at risk of trafficking as migration opportunities can easily lure them... if parents want to report, it is difficult to maintain the confidentiality of the issue among children or the whole community; this discourages them from reporting.*

It was disturbing to find that the girl children faced harassment in most instances, as observed by most participants. Moreover, they mentioned that the majority of the reports from camp parents were regarding missing children, and a few were for girls facing harassment. They also added that the local leaders, frequently called the Majhi, generally indirectly discouraged reporting and gave hope that they would solve the case through the community members.

## Discussion

The issue of child labor among Rohingya and host community children is deeply intertwined with socioeconomic factors, particularly poverty, and was impacted by the COVID-19 pandemic. Interviews conducted with working children aged 11–18 years reveal a troubling landscape where economic necessity drives both Rohingya and host community families to engage their children in labor, often at the expense of education and well-being. A few studies mentioned similar findings that poverty significantly impacts children's education and well-being, compelling families to depend on child labor for survival [19,20].

Poverty emerges as the predominant factor compelling families to send their children to work. In the host community, decisions regarding child labor are often made by fathers or jointly by parents, with many families citing the death of a parent or illness as critical triggers for this decision. Similarly, Rohingya children frequently reported being compelled to work following a wage earner's illness during the pandemic, highlighting how economic instability can lead to increased child labor participation. In many cases, decisions regarding child employment are made by parents, particularly fathers, who may see no alternative due to financial strain. A similar finding was observed in a study from Thailand [21]. The loss of a wage earner due to illness or death further exacerbates these challenges, pushing children into the workforce at a young age. For instance, many Rohingya children reported starting work to support their families after a primary breadwinner fell ill during the pandemic.

The interviews indicated that many children had been engaged in labor for extended periods, ranging from weeks to several years, which underscores the normalization of child labor in these communities. For instance, while host children typically drop out of school due to a lack of time caused by work commitments, Rohingya children often cited limited educational opportunities as a reason for their withdrawal from schooling. A study conducted in Bangladesh perspective also mentioned this issue [6].

The COVID-19 pandemic significantly altered the landscape of child labor. During lockdowns, many children could not work outside their homes, leading to a temporary decrease in employment opportunities. However, as restrictions eased, there was a notable resurgence in child labor activities. Children began engaging in various jobs such as assisting in shops and restaurants or carrying heavy packages for payment. This shift indicates that while the pandemic initially disrupted child labor patterns, it also reinforced existing vulnerabilities and economic pressures.

Moreover, the pandemic exacerbated existing issues such as trauma and familial disputes among children. Many articles mentioned that the COVID-19 pandemic has had far-reaching impacts on various aspects of society, including child labor [22–24].

The types of work performed by these children vary significantly between the host and Rohingya communities. Host community children often engage in hazardous occupations like welding and electrical work, while Rohingya children are typically involved in physically demanding tasks such as carrying heavy items or assisting with construction. The physical toll of such labor is concerning; many children reported experiencing pain from their work conditions, which could have long-term health implications.

Additionally, gender dynamics play a role in child labor practices. While boys are more frequently sent to work outside the home, girls in both communities are often restricted from engaging in labor due to cultural norms. However, those who do participate tend to be involved in domestic work or local NGO activities. This gender segregation reflects broader societal attitudes towards education and employment opportunities for girls. The gender dynamics related to child labor in Rohingya and host communities are distinct from other regions [25–27].

Addressing child labor within these communities is fraught with challenges. Many parents view child labor as a necessary contribution to household income rather than an issue requiring intervention. This perspective is compounded by fears regarding reporting abuses or seeking help due to potential retaliation from armed groups operating within or near refugee camps.

Furthermore, a lack of awareness about legal protections and support services inhibits effective reporting and intervention strategies. Stakeholders noted that while there is a willingness among parents to report issues related to child protection, concerns about confidentiality and repercussions deter them from taking action. Similar perceptions were found in another article [28].

The results of the study on child labor in the Rohingya and host communities offer valuable insights, but they also have certain limitations that should be considered. A limitation of this study is its reliance on data collected in 2022, which may not fully reflect current conditions. Given the time elapsed, a more long-term study is necessary to substantiate claims about child labor in these communities. Additionally, the rapid qualitative assessment lacked parents' observations, which could have provided deeper contextual insights. The data were based on interviews with children and key informants, which may introduce biases due to social desirability or recall bias. While the study provides insights into the immediate effects of child labor and the COVID-19 pandemic, it may not fully capture the long-term consequences on children's health, education, and well-being. The study does not account for other concurrent factors, such as sex labor, changes in humanitarian aid or shifts in local economies, which may have also influenced the dynamics of child labor. Building on this qualitative finding, a quantitative study is needed to assess the proportion of children engaged in hazardous work, disaggregated by age, gender, and type of employment. Moreover, a collaboration is needed with international organizations to align local efforts with global frameworks like Sustainable Development Goal (SDG) 8.7, which aims to eradicate child labor by 2025.

## Conclusions

This qualitative study on child labor, well-being, and the impact of COVID-19 among Rohingya and host community children in Bangladesh reveals a complex interplay of factors contributing to their vulnerability. Poverty, lack of educational opportunities, and economic hardships within families are primary drivers of child labor in both communities. The COVID-19 pandemic exacerbated these challenges, leading to increased child labor and exposure to hazardous work conditions. The study highlights significant gender disparities in child labor, with boys being more likely to engage in hazardous occupations. Additionally, cultural norms and societal expectations often dictate the roles of girls and boys, limiting their educational and occupational opportunities. While there are efforts to address child protection issues, challenges such as a lack of confidentiality, fear of retaliation, and limited access to reporting mechanisms hinder their effectiveness. To improve the situation, it is imperative to implement comprehensive strategies that address the root causes of child labor, provide access to education and vocational training, and strengthen child protection systems. Moreover, fostering community awareness and promoting gender equality are crucial steps towards protecting the rights and well-being of children in these vulnerable communities.

## Supporting information

**S1 Text. Questionnaire for Evaluation of Child Labor Situation in The Context of Rohingya And Host Community in Cox's Bazar, Bangladesh.**
(PDF)

## Author contributions

**Conceptualization:** Ahmed Hossain.

**Data curation:** Ahmed Hossain.

**Formal analysis:** Ahmed Hossain.

**Investigation:** Mohamad Alameddine.

**Methodology:** Ahmed Hossain, Heba Hijazi, Mohamad Alameddine.

**Resources:** Ahmed Hossain, Mohammad Ali.

**Software:** Ahmed Hossain.

**Supervision:** Ahmed Hossain.

**Validation:** Heba Hijazi.

**Writing – original draft:** Ahmed Hossain.

**Writing – review & editing:** Ahmed Hossain, Mohammad Ali, Heba Hijazi, Mohamad Alameddine.

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
