## [Decision Letter · Decision Letter 0]

PGPH-D-24-02286

Vulnerabilities of Rohingya and Host Community Children in Bangladesh: A Qualitative Study on Child Labor, Well-being, and the Impact of COVID-19

Dear Dr.Hossain,

Thank you for submitting your manuscript to PLOS Global Public Health. After careful consideration, we feel that it has merit but does not fully meet PLOS Global Public Health’s publication criteria as it currently stands. Therefore, we invite you to submit a revised version of the manuscript that addresses the points raised during the review process.

EDITOR: After careful review of the manuscript by two reviewers, it it needs major revision. I would suggest to address the comments from the reviewers and submit the revised version of the manuscript to be considered for publication.

We look forward to receiving your revised manuscript.

Kind regards,

Mohammad Mainul Islam, Ph.D.

Academic Editor

Journal Requirements:

1. Thank you for uploading your study's underlying data set. Unfortunately, the repository you have noted in your Data Availability statement does not qualify as an acceptable data repository according to PLOS's standards.

2. Please provide an Author Summary. This should appear in your manuscript between the Abstract (if applicable) and the Introduction, and should be 150–200 words long. The aim should be to make your findings accessible to a wide audience that includes both scientists and non-scientists. Sample summaries can be found on our website under Submission Guidelines:

https://journals.plos.org/globalpublichealth/s/submission-guidelines#loc-parts-of-a-submission

Additional Editor Comments (if provided):

Two reviewers carefully reviewed the manuscript, and they concluded that it needed major revision. I suggest addressing the reviewers' comments and submitting the revised version to be considered for publication.

Reviewers' comments:

Reviewer's Responses to Questions

**Comments to the Author**

1. Does this manuscript meet PLOS Global Public Health’s publication criteria?

Reviewer #1: Yes

Reviewer #2: Partly

2. Has the statistical analysis been performed appropriately and rigorously?

Reviewer #1: N/A

Reviewer #2: I don't know

3. Have the authors made all data underlying the findings in their manuscript fully available (please refer to the Data Availability Statement at the start of the manuscript PDF file)?

Reviewer #1: Yes

Reviewer #2: No

4. Is the manuscript presented in an intelligible fashion and written in standard English?

Reviewer #1: Yes

Reviewer #2: Yes

Reviewer #1: Thank you for the opportunity to review this paper on child labor and the Rohingya and Host Community Children in Bangladesh. The study is important and its findings highlight the various challenges that policymakers and public health workers face in improving the conditions of children in this area. The qualitative data is impressive in the access and tact that it took to secure interviews with children and key informants. Below, I have some suggestions, that I hope will help improve this manuscript.

In the Abstract, in the Methods section, authors mentioned that they conducted a “qualitative survey”? I think the description of the study would be better put as a “qualitative study comprised of in-depth interviews with a convenient and purposive sample”

In the Abstract, in the Conclusion section: Authors write, “The study found that Rohingya and host community children are increasingly involved in child labor”. There may be a *perception* of increased child labor, but as this is a qualitative study, we cannot make arguments and claims of representation here and quantitative increase of child labor for this population. Rather, you were able to unearth the varied reason for child labor participation in the Rohingya context and as a response to Covid-19.

On P. 4: Author writes, “This creates a cycle where education is deprioritized, and children are exposed to unsafe and exploitative labor conditions, further entrenching their families in poverty”. But I am a little unclear here how unsafe and exploitative labor conditions lead to further poverty. Perhaps a slight clarification here.

On P. 5-6: Author writes, “In this process, 20 in-depth interviews and 20 key-informant interviews (KIIs) were conducted.” And then later, “In the qualitative study, we performed twenty in-depth interviews (IDIs) and twelve KIIs”. I am a bit confused by the different numbers here—a typo perhaps?

On P. 7: I appreciate the care that the authors took to respect the privacy and autonomy of children interviewed for this project (e.g. not audio-recording). They write, “the underage participant’s available guardian, either their parents or the business owner, verbally agreed to their involvement”. Was the decision to seek approval from business owners/employers of these children, rather than their parents because of logistical reasons? In some contexts, child assent and parental consent are not interchangeable with the business owners who may exercise power over the children.

On P. 7: Authors write that “They were involved in diverse jobs”. How do these jobs accord with the definitions and typology of child labor discussed on P. 5? How many were economically active, child labor, or hazardous work?

On P. 7, for the Findings as a whole, I have three major suggestions. (1) I would have liked to have seen a little bit more representative quotes from respondents interwoven in the data analysis. There are some that are already in the text, but some more quotes if space allows, would help bring in the voices of those interviewed. (2) While the sample is on the smaller size (which is fine!), the Findings would also benefit from statements such as, “Of the 20 children interviewed, 5 mentioned that they were working because of X, 7 mentioned that they were working because of Y, etc.” This gives us a sense of the spread. (3) Just a little bit more interpretive work is needed to help readers understand the implications of each quote and finding. For example, on P. 9, authors write “children were also seen working outside the camps as drivers”. This should be pointed out as a hazardous activity.

On P. 8: Authors mention, “The children principally mentioned two problems int heir working places: painful, tedious work and untimely, partial, or no payment”. This I think merits more elaboration, as these point to the risks and potential exploitation of these children by employers.

On P. 10, the section on Health and Wellbeing seems a bit short. Given that we are interested in the global health implications of child labor, this section should elaborate more.

On P 15, in the Discussion section, the authors do a good job in discussing the various implications of their findings. They also point to the limitations of this exploratory study. It would be helpful to include a few sentences about future directions of this research, or lines of inquiry that they wish other researchers to pursue.

Reviewer #2: Thank you very much for the opportunity to review this manuscript. It deals with the topic of the paramount importance--which is child labor in refugee camps for displaced Rohingya people--explored qualitatively in Cox’s Bazar District of Bangladesh. While the article presents important findings, it also falls short on several aspects:

First, the introduction, in my opinion, lacks depth. The question of child labor as a subject of academic and health research is not sufficiently introduced.

Second, in the methods authors detail the exact locations of the informants’ communities, going as far as listing the latitudes & longitudes. This is questionable as it creates a possibility of passive identification of informants. I do understand that anonymity is a luxury especially in a community such as a refugee camp, but it is important to adhere to the best practices as there could be a scenario of this publication, if published, endangering the research subjects. Furthermore, authors don’t explain why they chose these exact communities. Authors use the URL link to a third-party repository for a reader to read about the design of this study, while the link itself is set to private, even after registering on this website.

Third, given the sensitivity of the topic, the description of ethical arrangements for this study was lacking—it is highly formal, in my opinion. We do not know whether children were compensated for their interviews, who conducted interviews, how recruitment happened, how consent happened and where, where the informants were interviewed (at work, at ‘home’ or elsewhere and how that might have affected the data generated) and how authors did their best to avoid coercion of informants given the power imbalances. What was the role of illiteracy in shaping the written consent practice? As such, are the people who conducted the interviews part of the authorship of this article? A clarity on authors’ roles and responsibilities would be a great assent in evaluating this article.

Furthermore, It is particularly striking that authors seek approval from the business owners, per quote:

‘’Respondents’ participation in the survey was entirely voluntary, and informed consent was obtained from each participant. Prior to participation, the underage participant's available guardian, either their parents or the business owner, verbally agreed to their involvement in a note taking interview (p10)’’

One may wonder how that was ‘entirely voluntary’ and with ‘no pressure’ given that the business owners—per the article itself— exploited the said children and directed violence I them? Why were business owners considered as consent- authorising guardians in the first place?

Fourth, the authors initially talk about the interviews, while they also refer to the data collection method as a survey. Clarity and consistency are needed.

Fifth, the article states that the data was collected in 2022. Since then, quite some time passed and to make an ambitious claim about the child labor in these communities at large a more long-term study is needed. Perhaps the authors should acknowledge the methodological limitations of the rapid qualitative assessment, especially as there were no participant observations conducted. Of course, we can’t expect the precarity of labor camps to be magically resolved, but the overall tone should be adjusted as the COVID-19 is over as far as the international public emergences go.

Sixth, and more conceptually, while authors make a conclusion that ‘Older boys (15-17 years) were more likely than girls to engage in paid work’. As such, I wonder if the authors probed for the highly sensitive topic of sex labor, as authors briefly touched upon the fact that ‘child marriage and polygamy tended to increase as coping strategies to alleviate the shortage of food’, which can be a euphemism for sex labor. Authors should clarify that topic as it could be a data collection limitation and misinterpreted findings. Here, for instance, is the report of UN acknowledging that this same community is affected by forced sex labor:

https://www.iom.int/news/girls-sold-forced-labour-largest-group-trafficking-victims-identified-iom-bangladesh-refugee-camps

While reading the article I kept thinking that the authors go somewhat soft on the business owners who are the direct beneficiaries of the labor value extracted from children. Perhaps a more critical stance is needed beyond simply highlighting the omnipresent poverty and cultural norms.

Finally, can authors provide the list of questions they asked to the informants, as an appendix?

Taken together, I believe that after the refinements this article would be a good contribution to the journal.

**Do you want your identity to be public for this peer review?** For information about this choice, including consent withdrawal, please see our Privacy Policy

Reviewer #1: No

Reviewer #2: No

---

## [Decision Letter · Decision Letter 1]

PGPH-D-24-02286R1

Vulnerabilities of Rohingya and Host Community Children in Bangladesh: A Qualitative Study on Child Labor, Well-being, and the Impact of COVID-19

Dear Dr. Hossain,

Thank you for submitting your manuscript to PLOS Global Public Health. After careful consideration, we feel that it has merit but does not fully meet PLOS Global Public Health’s publication criteria as it currently stands. Therefore, we invite you to submit a revised version of the manuscript that addresses the points raised during the review process.

Two reviewers have provided their thoughts on your revised manuscript. Reviewer 2 has a minor suggestion to improve the introduction to your study. Please carefully consider their suggestion and revise your manuscript accordingly.

We look forward to receiving your revised manuscript.

Kind regards,

Sarah Jose, Ph.D.

Staff Editor

Journal Requirements:

Additional Editor Comments (if provided):

Reviewers' comments:

Reviewer's Responses to Questions

**Comments to the Author**

Reviewer #1: All comments have been addressed

Reviewer #3: (No Response)

publication criteria?

Reviewer #1: Yes

Reviewer #3: Yes

3. Has the statistical analysis been performed appropriately and rigorously?

Reviewer #1: N/A

Reviewer #3: I don't know

4. Have the authors made all data underlying the findings in their manuscript fully available (please refer to the Data Availability Statement at the start of the manuscript PDF file)?

Reviewer #1: No

Reviewer #3: Yes

5. Is the manuscript presented in an intelligible fashion and written in standard English?

Reviewer #1: Yes

Reviewer #3: Yes

Reviewer #1: I applaud the authors for this major revision of their manuscript. They have sufficiently addressed my concerns raised in the previous round of review.

Reviewer #3: It is a nice article. But definition of child labour, legal frame works in the Bangaladesh should be mentioned.

**Do you want your identity to be public for this peer review?** For information about this choice, including consent withdrawal, please see our Privacy Policy

Reviewer #1: No

Reviewer #3: **Yes: ** Dr Samir Kumar Nanda

---

## [Decision Letter · Decision Letter 2]

Vulnerabilities of Rohingya and Host Community Children in Bangladesh: A Qualitative Study on Child Labor, Well-being, and the Impact of COVID-19

PGPH-D-24-02286R2

Dear Dr. Hossain,

We are pleased to inform you that your manuscript 'Vulnerabilities of Rohingya and Host Community Children in Bangladesh: A Qualitative Study on Child Labor, Well-being, and the Impact of COVID-19' has been provisionally accepted for publication in PLOS Global Public Health.

Best regards,

Julia Robinson

Executive Editor

Reviewer Comments (if any, and for reference):

Reviewer's Responses to Questions

**Comments to the Author**

Reviewer #1: All comments have been addressed

Reviewer #3: All comments have been addressed

publication criteria?

Reviewer #1: Yes

Reviewer #3: Yes

3. Has the statistical analysis been performed appropriately and rigorously?

Reviewer #1: N/A

Reviewer #3: Yes

4. Have the authors made all data underlying the findings in their manuscript fully available (please refer to the Data Availability Statement at the start of the manuscript PDF file)?

Reviewer #1: No

Reviewer #3: Yes

5. Is the manuscript presented in an intelligible fashion and written in standard English?

Reviewer #1: Yes

Reviewer #3: Yes

Reviewer #1: N/A

Reviewer #3: No comments

**Do you want your identity to be public for this peer review?** For information about this choice, including consent withdrawal, please see our Privacy Policy

Reviewer #1: No

Reviewer #3: **Yes: ** Dr Samir Kumar Nanda
